# SpaLLM: Unified Compressive Adaptation of Large Language Models with Sketching

## Abstract

Compressive adaptation approaches, such as QLoRA, are widely popular alternatives for reducing memory requirements during fine-tuning of large language models (LLMs) while producing models capable of handling various downstream tasks. The key idea is to employ a "two-tower" architecture: compressing pretrained LLM parameters into compact representations and fine-tuning the additive full-precision adapter, which typically has few tunable parameters in low-rank format. However, the strict algebraic assumptions, such as low-rank assumption, and the complexity of composing two-tower architectures are some of the known shortcomings, resulting in a poor accuracy-efficiency trade-off. In response to these known limitations, we propose SpaLLM (Sketched Parameter Adaptation of LLMs), a novel compressive adaptation approach for LLMs. This method is also the first to illustrate parameter-sharing compression methods for LLM finetuning, which, unlike QLoRA, are free from strict low-rank algebraic assumptions on adapters. Furthermore, our proposal unifies model compression and adaptation into a single, streamlined process, eliminating the need for two-tower architectures. SpaLLM sketches pre-trained LLM weights into lookup tables and directly fine-tunes the values in these tables. This approach simplifies LLMs' compressive adaptation workflow, potentially improves multi-user serving efficiency, and delivers significantly better accuracy for both natural language understanding and generation tasks. Moreover, by avoiding the "two-tower" architecture, our framework only requires one compressed matrix multiplication per layer during inference, demonstrating superior inference efficiency compared to previous methods.

## 1 Introduction

Recent advancements in Large Language Models (LLMs) have demonstrated exceptional performance in Natural Language Processing (NLP), enabling a broad spectrum of downstream applications. LLMs have demonstrated impressive generalization abilities across many downstream tasks in a zero-shot manner. However, compared to training-free methods such as in-context learning (Dong et al., 2022; Rubin et al., 2021) and few-shot prompting (Brown, 2020; Song et al., 2023), fine-tuning on these LLMs is often the ideal method to achieve optimal performance on a specific downstream task (Ding et al., 2023). Clearly, full-precision fine-tuning on these LLMs are often impractical due to the massive requirement of high-performance computing devices such as GPUs. As a result, Parameter-Efficient Fine-Tuning methods (PEFT), such as Low-Rank Adaptation (LoRA) (Hu et al., 2022), emerged as a less resource-intensive approach to fine-tuning while achieving reasonable accuracy in NLP applications. Clearly, there is a trade-off between accuracy and efficiency. Nevertheless, these PEFT methods still heavily relies on significant compute and memory resources, especially when the base model is large, due to their "two-tower" architecture.

**Compressive adaptation of LLMs.** This paper studies the compressive adaptation of LLM, where the limited hardware resources cannot even afford the full-precision storage of LLM parameters. In this case, we have to compress LLM parameters to lower precision, and then perform adaptation. PEFT on top of a compressed model has become a popular compressive adaptation approach to address the high resource expense (Dettmers et al., 2023; Li et al., 2023; Liu et al., 2023; 2024b; Qin et al., 2024; Xu et al., 2023) associated with LLM adaptations. Most of these approaches all use variations of LoRA-based adapters to perform the fine-tuning. For example, QLoRA (Dettmers et al., 2023) first compress a model to 4-bit precision, then add a set of LoRA adapters that are fine-tuned

by back-propagating through the compressed weights. LoftQ (Li et al., 2023) finds an initialization for the LoRA adapters that approximates the full-precision parameters before quantization, thus reducing the performance gap.

**Challenges of two-tower compressive adaptation.** Two-tower approaches with compress-finetune strategies face two challenges: 1) **Strict algebraic assumption:** The existing methods for compressive adaptation are built on strict algebraic assumptions about the adapters Dong et al. (2022); Liu et al. (2023). For instance, all the LoRA-based compressive adaptation approaches assume that *the difference between fine-tuned model parameters and base model parameters forms a low-rank matrix*. However, studies have shown that the difference in weights between fully fine-tuned and base parameters can be high-rank (Liu et al., 2024a). Additionally, the compression process is lossy, potentially requiring the adapters to exert extra effort to compensate for the loss. As a result, two-tower methods underperform when base parameters are compressed to lower bit levels, such as 3-bits or fewer (Yin et al., 2023). 2) **Difficulty in implementation:** The two-tower compressive adaptation approaches demand complex implementation during both training and inference. If not carefully managed, the distinct two-path structure introduces additional overhead in both computation and memory usage. For example, in the QLora formulation, the first path dequantizes low-bit pre-trained weights and uses them to perform matrix multiplication with the input. In the second path, the input undergoes matrix multiplication with full-precision low-rank adapter parameter matrices. The results from both paths are then combined to produce the final output. Since these paths involve multiplications at different precision levels, they must be handled as separate operations, requiring significant system-level optimizations. As a result, finding methods that mitigate the two problems mentioned above and provide better efficiency-accuracy trade-offs for LLM fine-tuning remains an active area of research.

**Our proposal: unified adaptation directly on compressed parameters.** This paper proposes a unified approach for fine-tuning large language models (LLMs) that is both parameter- and memory-efficient. The proposed method fine-tunes directly on compressed model parameters, without adhering to strict algebraic assumptions. This flexibility allows for improved model performance, especially when heavy compression is needed to meet hardware resource constraints. By applying matrix multiplication-based updates directly on the compressed parameters, the approach circumvents the need for extensive system-level optimizations, such as quantization and dequantization. This not only simplifies the process but also reduces computational costs by eliminating the need for dequantization and the use of complex "two-tower" architectures.

**SpaLLM: Sketched Parameter Adaptation of LLMs.** In this paper, we propose SpaLLM, denoted as Sketched Parameter Adaptation of LLMs, a novel compressive adaptation approach for LLMs. Our method is the first to apply sketching-based parameter-sharing techniques to LLM fine-tuning without relying on algebraic assumptions, which are common in methods like QLoRA and LoftQ. SpaLLM streamlines the process by combining model compression and adaptation into a single workflow, eliminating the need for complex two-tower architectures. SpaLLM involves transforming pre-trained LLM weights into lookup tables and directly fine-tuning these values, simplifying the adaptation process. This not only enhances the efficiency of serving multiple users but also improves accuracy for both natural language understanding and generation tasks. Additionally, by avoiding the two-tower structure, SpaLLM requires just one compressed matrix multiplication per layer during inference, significantly boosting inference efficiency compared to previous methods.

## 2 SpaLLM: Unified Compressive Adaptation

In this section, we introduce our method, SpaLLM (Sketched Parameter Adaptation of LLMs), emphasizing how sketching algorithms facilitate parameter sharing. This approach enables scalable and unified compressive adaptation of LLMs.

### 2.1 Parameter Sharing as Unified Compressive Adaptation.

Parameter sharing involves techniques that allocate a fixed set of ML model parameters and reuse them across various parts of the model. For example, in LLMs, this often involves sharing weights between various functional components. A common, though simplistic, form of parameter sharing in LLMs is group query attention (GQA) (Ainslie et al., 2023), where different attention heads with

distinct key and value projections share the same query projections. The parameter sharing approach is particularly useful in LLMs, where parameter redundancy can quickly lead to increased memory and computational demands. In this work, we propose that beyond improving efficiency, parameter sharing also enables a unified, compressive adaptation of LLMs across different downstream tasks.

**Parameter sharing for more compression.** In the context of model compression, parameter sharing offers an effective way to reduce model size with minimal impact on performance. While traditional structural parameter sharing methods like GQA are widely used, recent research highlights the advantages of element-wise parameter sharing (Chen et al., 2015; Desai & Shrivastava, 2022; Desai et al., 2022; 2023; Desai & Shrivastava, 2023), which goes beyond conventional techniques like pruning (Frantar & Alistarh, 2023; Ma et al., 2023; Zhou et al., 2024) and quantization (Frantar et al., 2022). In this approach, model parameters are categorized into two types: *virtual parameters* and *trainable parameters*. Virtual parameters correspond to specific locations within the model architecture where a parameter is required. Trainable parameters, on the other hand, represent the values that are updated during the training process. The number of trainable parameters is significantly smaller than the number of virtual parameters. Element-wise parameter sharing maps each virtual parameter to a corresponding trainable parameter, with multiple virtual parameters often sharing the same trainable parameter. During training, gradients are first calculated for each virtual parameter. These updates are then aggregated for the shared trainable parameters, which may receive gradients from several virtual parameters. The trainable parameters are updated by averaging these gradients. Empirical evidence suggests that this element-wise parameter-sharing approach opens up new possibilities for compressing machine learning models effectively.

**Parameter sharing as regularization.** Beyond compression, parameter sharing can also serve as a form of regularization in post-training processing of LLMs. For a pre-trained LLM, we treat all parameters as virtual and map each virtual parameter to a trainable one. During this process, a single trainable parameter may receive values from multiple virtual parameters, and these values are averaged. This averaging acts as a regularization technique for the original parameters of the LLM. By requiring different virtual parameters to rely on the same trainable parameter, the model is constrained, reducing over-fitting by limiting its flexibility (Shakerinava et al., 2024). This ensures that the shared parameters capture more general, robust information from the training data.

**Regularize, then adapt: parameter sharing for unified compressive adaptation.** We propose parameter sharing as a valuable method that combines the benefits of compression and regularization. Starting with a pre-trained LLM, we apply parameter sharing to compress the model while simultaneously introducing regularization. Following this, we fine-tune the trainable parameters while keeping the mapping between virtual and trainable parameters fixed, maintaining the regularization effect during fine-tuning. This integrated approach not only reduces the model's size but also allows for task-specific fine-tuning, ensuring the model remains both efficient and adaptable. It has proven effective in reducing the memory footprint of LLMs while preserving their generalization ability, making it a practical solution for deploying large models in environments with limited resources.

## 2.2 SKETCHING FOR PARAMETER SHARING

We propose sketching algorithms to enable efficient parameter sharing for unified compressive adaptation of LLMs. Our approach involves three key steps. First, we apply row-wise parameter sketching, where each row of the model parameters is compressed into sketched parameters. Next, we introduce a weighted version of Lloyd's algorithm to optimize this sketching process. Finally, we implement lookup tables to store the sketched parameters and their corresponding mappings, enabling efficient computation.

**Row-wise parameter sketching.** We perform parameter sketching for each row of the parameter matrix independently. As shown in Figure 1, given a parameter matrix $\Theta \in \mathbb{R}^{n \times d}$ in the LLM, we perform parameter sketching in every row $\theta \in \mathbb{R}^d$ of $\Theta$. Formally, we approximate $\theta$ as:

$$\hat{\theta} = \Pi w, \tag{1}$$

where $w \in \mathbb{R}^k$ is the sketched parameter and $\Pi \in \mathbb{R}^{d \times k}$ is a one-hot sketching matrix. Every row of $\Pi$ is a one-hot vector with only one nonzero value as 1. In this formulation, every entry in $\theta$ is mapped to an entry in $w$, and multiple entries in $\theta$ can be mapped to the same entry in $w$. We note that $k < d$ so we compress the $\theta$ with one-hot mapping $\Pi$ and sketched parameter $w$.

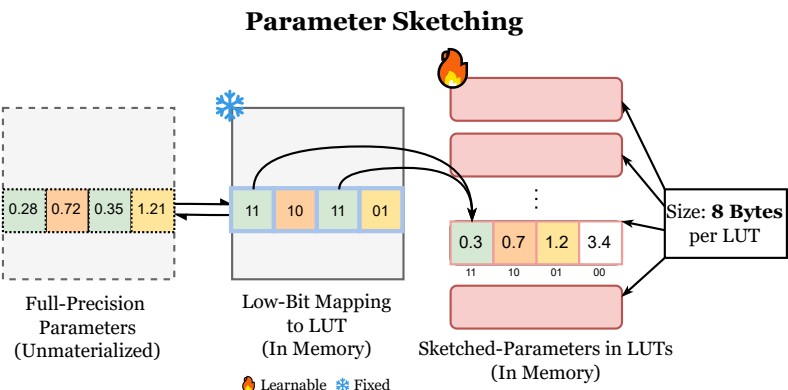

Figure 1: Illustration of parameter sketching. We use 2-bit sketching as an example.

**Weighted Lloyd's algorithm for learning sketched parameters.** Common sketching methods, such as Bloom filters (Bloom, 1970) and Count Min Sketch (Cormode & Muthukrishnan, 2005), often employ randomized hashing as the sketching matrix, to minimize the impact of collisions on the sketching quality. However, since LLMs parameters encapsulate rich pre-trained knowledge and are sensitive to perturbations (Frantar & Alistarh, 2023), randomized sketching would greatly degrade the quality of the model. Moreover, a randomized-hashing-based sketching is good for preserving heavy hitters, but LLM parameters do not exhibit heavy-hitter patterns (Xiao et al., 2023). Therefore, we propose to learn the sketch by taking inspirations from existing quantization approaches (Zhang & Shrivastava, 2024). We perform Lloyd's algorithm (Lloyd, 1982) to learn a set of $k$ centroids $w \in \mathbb{R}^k$ for the row of parameters $\theta$, which are inversely weighted by the Hessian diagonals $\mathrm{diag}(\mathbf{H}^{-1}) = \mathrm{diag}\left((\mathbf{X}\mathbf{X}^\top)^{-1}\right)$. Here, $\mathbf{X} \in \mathbb{R}^{s \times d}$ is the sample input matrix to the layer, whose inner product with $\theta$ is the partial layer output, with $s$ being the sample size. The learned centroids $w$ then become the sketched parameters of the parameter row $\theta$.

**Learning the sketching matrix.** Once the centroids $w$ have been established, we learn the sketching matrix by employing the iterative loss-error-based quantization framework (Zhang & Shrivastava, 2024). We iteratively round each parameter in $\theta$ to the nearest centroid in $w$, while updating the not-yet-rounded parameters in $\theta$ to compensate for errors introduced by rounding according to the update rule in Frantar et al. (2022). This process maps each parameter in $\theta$ to a single entry in the sketched parameters $w$. As a result, the mapping becomes our sketching matrix $\Pi$, which is row-wise one-hot.

**Scaling up learning power by increasing sketched parameters.** As we increase the number of sketched parameters, the learning power of the adapted LLM increases. One potential way of increasing the number of sketched parameters is to increase the size of the centroids $w$. However, this would increase the size of the sparse sketching matrix $\Pi$. We instead increase the number of sketches by dividing each row of parameters into contiguous groups, where each group keeps its own sketch. This keeps the size of sketching matrices $\Pi$ constant while increasing the number of sketched parameters. We use groups per row (GPR) to scale up the learning power of our compressive adaptation method.

## 2.3 SCALING UP COMPRESSIVE ADAPTATION BY FINE-TUNING SKETCHED PARAMETERS

We propose to fix the mapping of the sketching algorithm ($\Pi$ in Eq equation 1) and only learn the floating-point sketched parameters $w$, allowing efficient adaptation to specific tasks while maintaining a compact LLM model.

**A unified compressive adapation paradigm.** As shown in Figure 2, unlike two-tower methods such as QLoRA, our approach utilizes a single-tower architecture, reducing complexity and computational overhead. By directly fine-tuning the sketched parameters while fixing the mappings introduced by the sketching matrix $\Pi$, we avoid the need for an additional external adapter with algebraic assumption, enabling more efficient parameter adaptation without sacrificing model performance.

**Adapting with a customized compression.** Our framework supports task-specific compression by offering flexibility in the sketching process. SpaLLM allows customization in setting the number of LUTs for each row, $\theta$, of the parameter, enabling the model to adjust based on input complexity and

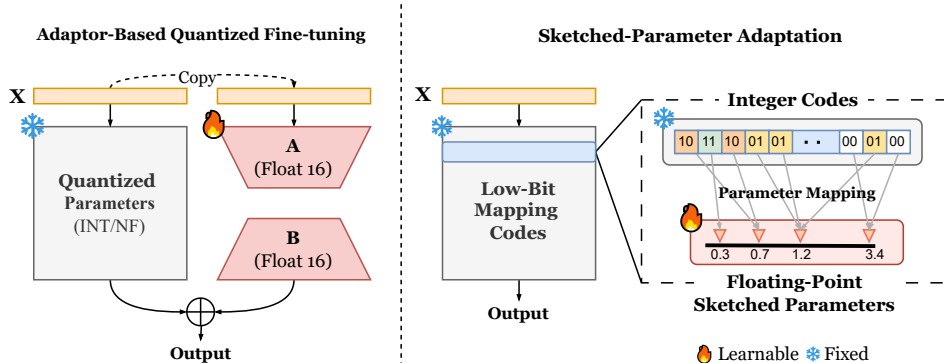

Figure 2: Comparison between two-tower adaptor-based architecture and our Sketched-Parameter Adaptation under low-bit compression. We use 2-bit sketching as an example.

available resources. This adaptability ensures optimal performance across various tasks. The fine-tuned, sketched parameters are tailored to the data's unique characteristics, allowing for personalized compression strategies that preserve essential information while minimizing redundancy.

**Natural support to dynamic maintenance of multiple adapters.** Our approach inherently supports the dynamic management of multiple adapters, enabling the model to switch between different compressed representations seamlessly. In SpaLLM, each adaptation builds multiple trainable LUTs. These LUTs have a unified format in both computation and memory storage. As a result, during both training and inference phases, SpaLLM supports multiple adapter as the update to LUTs are independent. This is particularly useful for multi-task learning, where different tasks might require varying degrees of compression. The ability to manage multiple sketched parameter in LUTs allows for efficient task switching without the need to fully retrain the model for each new task. Additionally, the framework ensures that memory overhead is minimized, as only the sketched parameters need to be stored and updated for each task, making it scalable for large-scale deployments.

## 2.4 SPALLM VS QLORA: RETAINS ALL NICENESS WHILE ELIMINATING THE LOW-RANK ASSUMPTION

From Figure 2, we can clearly see that both QLoRA and SpaLLM are parameter-efficient, with the number of trainable parameters being significantly smaller compared to the total number of parameters. This allows for efficient storage and processing on the adapters. Furthermore, like QLoRA, SpaLLM only requires memory equivalent to the quantized weights. This is especially critical when fine-tuning large LLMs, where parameter memory dominates overall memory requirements. As a result, SpaLLM preserves all the known advantages of QLoRA.

On the advantage side, SpaLLM makes no low-rank assumptions and can even handle full-rank updates. Many full-rank matrices can be compressed using the shared-parameter scheme without losing information. A simple identity matrix offers a perfect illustration. An identity matrix is full rank and can be compressed perfectly with only two values (0 and 1) using shared-parameter compression. At the same time, it is well known that there is no effective low-rank approximation for an identity matrix; thus, enforcing a low-rank assumption can lead to information loss compared to a shared-parameter assumption. In light of experimental results presented in(Liu et al., 2024a) establishing that the weight differences between fully fine-tuned and base parameters tend to be high-rank, shared-parameter adaptation seems like a more natural assumption for enabling PEFT.

## 3 EXPERIMENTS

We thoroughly evaluated the accuracy and efficiency of our proposed SpaLLM. Below, we describe our experimental setups. Please see Appendix B for detailed training and evaluation setups.

**Models**. We apply SpaLLM to several foundation models, including LLaMA-7B (Touvron et al., 2023a), LLaMA-2-7B and LLaMA-2-13B (Touvron et al., 2023b), LLaMA-3-8B and LLaMA-3-70B (Dubey et al., 2024).

**Datasets**. We train SpaLLM and baselines on the datasets WikiText-2 (Merity et al., 2016), GSM8K (Cobbe et al., 2021), and Alpaca (Taori et al., 2023). We evaluate the finetuned models on the test set of WikiText-2, GSM8K, and the CommonsenseQA benchmarks, including PIQA (Bisk et al., 2020), WinoGrande (Sakaguchi et al., 2019), HellaSwag (Zellers et al., 2019), OpenBookQA (Mihaylov et al., 2018), BoolQ (Clark et al., 2019), and ARC (Clark et al., 2018). In addition, we evaluate our model's generation quality when fine-tuned on the Alpaca dataset (Taori et al., 2023) using CommonsenseQA benchmarks and the LLM GPT-4o (2024-05-13) as a judge (Zheng et al., 2023). For all experiments that used the Alpaca dataset for fine-tuning and evaluation, we used the prompts provided by the official Alpaca codebase (Taori et al., 2023) to format the inputs for consistent comparison. Please see Appendix A for model and dataset details.

**Baseline Methods**. We compared SpaLLM against various LoRA-based compressive adaptation methods, including LoftQ (Li et al., 2023), QLoRA (Dettmers et al., 2023), QA-LoRA (Xu et al., 2023), and IR-LoRA (Qin et al., 2024). For our model that was fine-tuned on LLaMA-3-70B with 4 bit compression rate, we compared with the LLaMA-3-70B base model in full precision, and the Falcon-40B-Instruct model (Almazrouei et al., 2023).

**Hardware Usage.** All models, except for the LLaMA-3-70B ones, were fine-tuned and evaluated on NVIDIA A100 GPUs with 40GB of memory. The LLaMA-3-70B fine-tuning was conducted on a single NVIDIA L40S-48GB GPU.

**LLM-as-a-judge Evaluation.** We adopted the FastChat framework (Zheng et al., 2023) for LLM-as-a-judge evaluations. To assess model generation ability, we generated model responses on instructions from the Alpaca dataset (Taori et al., 2023) and Vicuna benchmark dataset (Chiang et al., 2023). Then, we prompted GPT-4o as a judge (Zheng et al., 2023) to compare the quality of model generations. For each test question, we provided GPT-4o with the question and corresponding generations from two different models (Model 1 and Model 2). The GPT-4o judge is prompted to compare the response twice: once with Model 1's response appearing first, and once with Model 1's response appearing second. This minimizes any bias introduced by the order of presentation. Specific generation and GPT-4o evaluation examples can be found in Appendix B.

### 3.1 MAIN RESULTS

**Benchmarks on WikiText-2 and GSM8K.** To compare SpaLLM against various compressive adaptation methods, we compressed LLaMA-2-7B and LLaMA-2-13B (Touvron et al., 2023b) using SpaLLM. We then fine-tuned and evaluated our model on WikiText-2 (Merity et al., 2016) and GSM8K (Cobbe et al., 2021), following experiment settings described by Li et al. (2023). For WikiText-2 evaluations, we report evaluation perplexity (lower is better). For GSM8K evaluations, we extract the final numerical result from model generations and calculate their accuracy against reference answers.

In Table 1, We compared our approach against established baselines, including QLoRA and LoftQ, two SOTA fine-tuning methods with model compression, as well as full-precision LoRA fine-tuning. On the WikiText-2 dataset, our method demonstrates better or comparable performance to QLoRA and LoftQ. Notably, when fine-tuned on the LLaMA-2-13B model, our approach outperforms baseline methods across the board. At a 3 bit compression rate, where achieving high performance is difficult, our method reached a perplexity of 5.05, even outperforming LoRA fine-tuned model with full-precision weights (5.12). SpaLLM also performs well on GSM8K evaluations: when fine-tuned on LLaMA-2-7B model, our method achieves over 3% higher precision than the baseline methods at both 4-bit and 2-bit compression. Remarkably, our method surpasses the accuracy of the full-precision LoRA model by 3.8% when fine-tuned on LLaMA-2-13B. These results displays the strong learning ability of our fine-tuning approach.

**LLM-as-a-judge comparison with LoftQ.** Additionally, we conducted a comparison between SpaLLM and LoftQ on language generation tasks, aiming to assess model performance as LLM assistants. For this task, we used the recently released LLaMA-3-8B (Dubey et al., 2024) as the base model. Both approaches were fine-tuned using 4,096 randomly sampled inputs from the Alpaca dataset (Taori et al., 2023), and evaluated on 300 test inputs randomly sampled from the same dataset. The training and test sets were verified to have no overlapping data. Table 2 presents the LLM-as-a-judge comparison on the model generations using GPT-4o as a judge. Out of the 300 test instructions, SpaLLM won the comparison 147 time, and tied with LoftQ 60 times, resulting in a

Table 1: LLaMA-2 model family evaluation results on the WikiText-2 and GSM8K datasets. In this context, GPR (Groups Per Row) refers to the number of sketched-parameter groups at each row, with higher values indicating a more granular sharing of parameters. The bit count for SpaLLM represents the number of bits used to encode the sketching matrices. We report evaluation perplexity (lower is better) for WikiText-2 and output accuracy for GSM8K. Instances where the model fails to converge are denoted as $N.A.$

| Method | #Bit | WikiText-2↓ | | Method | #Bit | GSM8K ↑ | |
| | | 7B | 13B | | | 7B | 13B |
|---|---|---|---|---|---|---|---|
| LoRA (r=64) | 16 | 5.08 | 5.12 | LoRA (r=64) | 16 | 36.9 | 43.1 |
| LoRA+Reg (r=64) | 16 | - | - | LoRA+Reg (r=64) | 16 | 34.4 | 45.3 |
| QLoRA (r=64) | 4 | 5.7 | 5.22 | QLoRA (r=64) | 4 | 35.1 | 39.9 |
| LoftQ (r=64) | 4 | **5.24** | 5.16 | LoftQ (r=64) | 4 | 35 | 45 |
| **SpaLLM (GPR=1)** | 4 | 5.32 | **4.81** | **SpaLLM (GPR=8)** | 4 | **38.4** | **49.1** |
| QLoRA (r=64) | 3 | 5.73 | 5.22 | QLoRA (r=64) | 3 | 32.1 | 40.7 |
| LoftQ (r=64) | 3 | **5.63** | 5.13 | LoftQ (r=64) | 3 | 32.9 | **44.4** |
| **SpaLLM (GPR=1)** | 3 | **5.63** | **5.05** | **SpaLLM (GPR=8)** | 3 | **33.1** | 43.9 |
| QLoRA (r=64) | 2 | N.A. | N.A. | QLoRA (r=64) | 2 | N.A. | N.A. |
| LoftQ (r=64) | 2 | 7.85 | 7.69 | LoftQ (r=64) | 2 | 20.9 | 25.4 |
| **SpaLLM (GPR=1)** | 2 | **7.40** | **6.22** | **SpaLLM (GPR=8)** | 2 | **23.7** | **33.8** |

Table 2: Model generation quality comparison using GPT-4o as a judge between SpaLLM and LoftQ

| Method | #Bit | Win | Loss | Tie | Win Rate | Loss Rate | Win-Loss Ratio |
|---|---|---|---|---|---|---|---|
| LoftQ (r=64) | 4 | 93 | 147 | 60 | 0.31 | 0.49 | 0.39 |
| **SpaLLM (GPR=8)** | 4 | **147** | **93** | **60** | **0.49** | **0.31** | **0.61** |

win-loss ratio of 0.61. This observation indicates that our model is more likely to cater to human preference when deployed as a LLM assistant in real-world settings.

**Benchmarks on the CommonsenseQA dataset.** We also present the 0-shot results from the CommonsenseQA benchmarks in Table 3. All the methods were fine-tuned on the LLaMA-7B (Touvron et al., 2023a). The LoRA-based adaptors all applied a rank of 64 and $alpha = 16$. For our method, we set up SpaLLM with one group per row in the compressed matrices, significantly reducing the model size. Indeed, our method only used 22 million trainable parameters. This small parameter budget still allowed SpaLLM to achieves the best average when compared to SOTA methods, consistent with the observations from the WikiText-2 and GSM8K experiments.

**Evaluations on larger models.** We extended the evaluation of our method to larger models, as shown in Table 4, where we present results for the recently released LLaMA-3-70B model. For fine-tuning, we utilized the entire Alpaca dataset (Taori et al., 2023) as training data and applied four groups of sketched-Parameter mappings per row. With a 4-bit compression rate, our method achieved better performance than the full-precision LLaMA-3-70B base model. Despite a modest 1% improvement in accuracy, SpaLLM reduced the memory footprint significantly, requiring only 39.8GB for inference.

Furthermore, we assessed the generation quality of our compressed model using the LLM-as-a-judge framework. Responses were generated based on questions from the Vicuna Bench dataset, and GPT-4o was prompted to compare the output of SpaLLM at 4-bit compression with that of Falcon-40B-Instruct (Almazrouei et al., 2023) at full precision. The results consistently favored our approach, with a win-loss ratio of 91%. Moreover, our 70B model, capable of fitting in a single NVIDIA L40S-48GB GPU, offers substantial efficiency gains compared to both the base LLaMA-3-70B and Falcon-40B-Instruct models, which require multiple GPUs for inference.

Table 3: Accuracy (%) comparison on 5 Commonsense QA datasets on the LLaMA-7B model

| Method | #Bit | #Params | CommonsenseQA | | | | | |
|---|---|---|---|---|---|---|---|---|
| | | | HellaSwag | PIQA | WinoGrande | ARC-e | ARC-c | AVG |
| LLaMA-7B | 16 | - | 56.3 | 78.2 | 67.1 | 67.3 | 38.2 | 61.4 |
| NormalFloat | 4 | 6.7B | 56.7 | 78.7 | 70.6 | 75.7 | 41.6 | 64.7 |
| QLoRA w/GPTQ | 4 | 160M | 57.4 | 77.6 | 66.2 | 70.9 | 41.8 | 62.8 |
| QLoRA | 4 | 160M | 61.8 | 78.1 | 68.4 | 75.8 | 43.6 | 65.5 |
| QA-LoRA | 4 | 160M | 58.6 | 78.0 | 66.9 | 71.2 | 43.9 | 63.7 |
| IR-QLoRA | 4 | 160M | 54.7 | 78.8 | 72.6 | 76.6 | 45.1 | 65.6 |
| **SpaLLM (GPR=1)** | 4 | 22M | 58.2 | 78.7 | 69.9 | 76.1 | 44.9 | **65.8** |

Table 4: LLaMA-3-70B experiment results: The top table presents the accuracy comparison (%) across seven Commonsense QA datasets, contrasting the full-precision LLaMA-3-70B base model with the version fine-tuned using SpaLLM. The bottom table shows the results from the LLM-as-a-judge evaluation, comparing the performance of the LLaMA-3-70B model fine-tuned with our method against the Falcon-40B-Instruct model.

| | | | CommonsenseQA Benchmarks | | | | | | | |
|---|---|---|---|---|---|---|---|---|---|---|
| Method | #Bit | Model Size | ARC-e | ARC-c | HellaSwag | PIQA | WinoGrande | OBQA | BoolQ | AVG |
| LLaMA-3-70B | 16 | 141.1GB | 0.87 | 0.60 | **0.66** | 0.83 | 0.80 | 0.38 | 0.85 | 0.71 |
| **SpaLLM (GPR=4)** | 4 | 39.8GB | **0.87** | **0.62** | **0.66** | **0.83** | **0.81** | **0.40** | **0.87** | **0.72** |

| | | | LLM-as-a-judge | | | | | |
|---|---|---|---|---|---|---|---|---|
| Method | #Bit | Model Size | Win | Loss | Tie | Win Rate | Loss Rate | Win-Loss Ratio |
| Falcon-40B-Instruct | 16 | 83.7GB | 6 | 59 | 15 | 0.08 | 0.74 | 0.09 |
| **SpaLLM (GPR=4)** | 4 | 39.8GB | **59** | **6** | **15** | **0.74** | **0.08** | **0.91** |

## 3.2 ABLATION STUDY

**Accuracy Ablation.** We evaluated the accuracy trend of SpaLLM at 4-bit precision to assess the impact of varying the number of sketched parameter groups on performance. As shown in Figure 3.2, we report results for configurations with 1, 2, 4, and 8 groups per row in the compressed weight matrix, alongside baseline comparisons with LoftQ (4-bit compression, $r = 64$) and LoRA (full precision, $r = 64$). The results indicate a clear improvement in accuracy as the number of groups increases, demonstrating that finer granularity in sketched parameter groups enhances the retention of information. On the LLaMA-2-7B model, SpaLLM surpasses the baselines at 8 groups per row and achieves comparable performance with just 4 groups per row, while using half the number of trainable parameters. Similarly, on the LLaMA-2-13B model, SpaLLM outperforms the baselines starting from 1 group per row, utilizing only 1/5 of the trainable parameters. These findings highlight the scalability of SpaLLM and its ability to efficiently utilize trainable parameters for knowledge retention during fine-tuning.

**Efficiency Ablation.** We compared the inference efficiency of SpaLLM with LoRA-based compressive adaptation methods, including QLoRA and LoftQ. To assess performance, we measured the time and memory usage required by each model to decode 128 tokens. For SpaLLM, we utilized kernels developed by Kim et al. (2024). The experiments were conducted on NVIDIA V100 GPUs (32GB memory). As shown in Table 5, SpaLLM achieves a 3× improvement in efficiency compared to LoRA-based methods, while consistently using less GPU memory than both QLoRA and LoftQ. This improvement is due to SpaLLM's ability to avoid the "two-tower" approach common in LoRA-based adaptations, resulting in lower memory usage and faster throughput by performing only a single pass during inference.

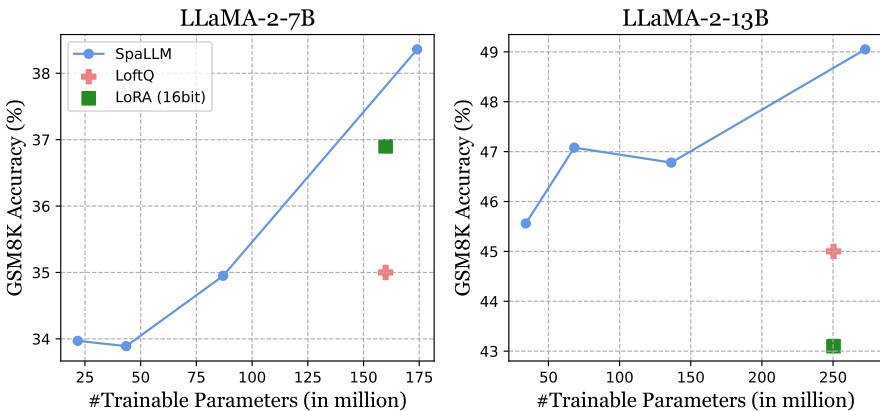

Figure 3: Accuracy Comparison on the GSM8K dataset under different trainable parameter budgets. The different number of trainable parameters are achieve by adjusting the number of groups of sketched parameter per row (GPR). On both plots, the groups per row used to create the four data points are 1, 2, 4, 8 (from left to right).

Table 5: Efficiency comparison between QLoRA and SpaLLM on LLaMA-2-7B and LLaMA-2-13B when decoding 128 tokens. Both QLoRA and LoftQ are included in the table, as their inference implementations are identical.

| Model | Method | #Bit | Time/Token (s) | Peak GPU Memory (GB) |
|---|---|---|---|---|
| LLaMA-2-7B | QLoRA/ LoftQ (r=64) | 4 | $0.19 \pm 0.02$ | 0.65 |
| | **SpaLLM (GPR=1)** | 4 | $\mathbf{0.06 \pm 0.01}$ | **0.58** |
| LLaMA-2-13B | QLoRA/ LoftQ (r=64) | 4 | $0.24 \pm 0.02$ | 1.15 |
| | **SpaLLM (GPR=1)** | 4 | $\mathbf{0.08 \pm 0.01}$ | **1.01** |

## 4 RELATED WORKS

**Compressive adaptation of LLMs.** The adaptation of LLMs to specific tasks while maintaining computational efficiency has been an active area of research. Parameter-Efficient Fine-Tuning (PEFT) methods have been created to mitigate the high cost of full fine-tuning on LLMs. Among these these methods, there has been two major tracks of work: low rank based adaptation methods, include LoRA (Hu et al., 2022), DoRA (Liu et al., 2024b); and structural sparse adaptation methods, including OFT (Qiu et al., 2024) and BOFT (Liu et al., 2023). Compressive adaptation methods aim to further reduce the memory and computational footprint of fine-tuning large models without sacrificing performance. Fine-tuning after quantization has become a popular approach. QLoRA (Dettmers et al., 2023) introduced the NormalFloat data type for performing quantization while fine-tuning additional LoRA adapters to mitigate the knowledge lost during quantization. LoftQ (Li et al., 2023) reduce the error between the compressed, fine-tuned model and the original full-precision model by approximating the base weights using the quantized weights and LoRA adapter during quantization. IR-LoRA (Qin et al., 2024). However, these existing compressive adaptation methods, due to their use of the two-tower approach (compressed base weights and adapters), face difficulties in managing complex implementation. Without careful management, the two distinct path during inference introduce additional overhead in compute and memory usage.

**Parameter Sharing in Machine Learning** Advances in machine learning have led to the exploration of parameter sharing as a means to reduce the memory footprint and computational costs typical of large models. HashedNets(Chen et al., 2015) applies compression before training by using a hash function to group weights into hashed buckets. Similarly, Desai et al. (2022) introduced ROBE, which compresses embedding tables through randomized hashing for more efficient storage and access. ROAST(Desai et al., 2023) extends these ideas by utilizing global weight sharing to improve both training and inference speed. STABLE-RPS (Desai & Shrivastava, 2023) refines

the ROAST method, introducing a gradient-scaling technique to enhance stability during training and improve model accuracy. However, applying these methods to LLMs poses challenges, as they rely on compression before training. This pre-training compression constrains the model's learning ability, which has led to limited success in applying parameter-sharing approaches to LLMs.

## 5 CONCLUSION

In this paper, we introduced SpaLLM, a novel compressive adaptation approach for LLMs that effectively addresses the limitations of existing methods like QLoRA. By eliminating the need for low-rank assumptions and unifying model compression and fine-tuning into a streamlined process, SpaLLM offers a more efficient solution for fine-tuning LLMs on resource-constrained hardware. The proposed sketching-based parameter-sharing mechanism allows SpaLLM to maintain high performance even under heavy compression, while significantly reducing the computational and memory overhead typically associated with two-tower architectures. Through extensive experiments, we demonstrated that SpaLLM not only outperforms state-of-the-art compressive adaptation methods in terms of accuracy but also achieves superior inference efficiency, making it an ideal choice for large-scale LLM deployments. Our approach's ability to adapt seamlessly across a variety of natural language understanding and generation tasks further highlights its versatility and robustness.

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

# A  MODELS AND DATASETS

We applied SpaLLM to fine-tune LLaMA-7B (Touvron et al., 2023a), LLaMA-2-7B, LLaMA-2-13B (Touvron et al., 2023b), LLaMA-3-8B and LLaMA-3-70B (Dubey et al., 2024). For dataset, we fine-tuned our model on GSM8K (Cobbe et al., 2021), WikiText-2 (Merity et al., 2016), and the Alpaca dataset (Taori et al., 2023). We show the details of each training dataset in Table 6.

We evaluated our model using GSM8K, WikiText-2, CommonsenseQA benchmarks, including PIQA (Bisk et al., 2020), WinoGrande (Sakaguchi et al., 2019), HellaSwag (Zellers et al., 2019), OpenBookQA (Mihaylov et al., 2018), BoolQ (Clark et al., 2019), and ARC (Clark et al., 2018), as well as LLM-as-a-judge (Zheng et al., 2023). For GSM8K, WikiText-2, and LLM-as-a-judge benchmarks, the model was assessed on language generation tasks, while CommonsenseQA involved multiple-choice tasks. We utilized the Language Model Evaluation Harness (lm-eval) (Gao et al., 2024) to conduct evaluations for the CommonsenseQA benchmarks.

Table 6: Details of the training datasets used to fine-tune SpaLLM

| Dataset | #Train | #Dev | #Test | Metrics |
|---------|--------|------|-------|---------|
| GSM8K | 7.47k | - | 1.32k | Accuracy |
| Wikitext-2 | 36.7k | 3.76k | 4.36k | Perplexity |
| Alpaca | 52k | - | - | - |

# B  EXPERIMENT SETTINGS

## B.1  FINE-TUNING ON LLaMA-2-7B AND LLaMA-2-13B

For all experiments, we tried learning rates from ($1 \times 10^{-4}$, $5 \times 10^{-5}$, $3 \times 10^{-5}$, $1 \times 10^{-5}$, $5 \times 10^{-6}$, $2 \times 10^{-6}$). We present the specific training setting in Table 7.

Table 7: Training settings for SpaLLM on the GSM8K and WikiText-2 dataset

| Dataset | Model | Learning Rate | Batch Size | Epochs |
|---------|-------|---------------|------------|--------|
| GSM8k | LLaMA-2-7B | $3 \times 10^{-5}$ | 4 | 10 |
|  | LLaMA-2-13B | $3 \times 10^{-5}$ | 4 | 10 |
| WikiText-2 | LLaMA-2-7B | $3 \times 10^{-5}$ | 4 | 10 |
|  | LLaMA-2-13B | $3 \times 10^{-5}$ | 4 | 10 |

## B.2  LLM-AS-A-JUDGE ON LLaMA-3-8B

We fine-tuned LLaMA-3-8B using LoftQ and SpaLLM. As there is only one split in the Alpaca dataset, we selected 4096 rows from the dataset as our training data, and a distinct set of 300 row from the dataset as our test data. We trained 3 epochs for both methods. Details on the training settings are presented in Table 8. In addition, we present an example of model generation along with the GPT-4 judgment in Table 9.

Table 8: Training settings for LoftQ and SpaLLM on 4096 samples from the Alpaca dataset

| Method | Model | Learning Rate | Batch Size | Epochs |
|--------|-------|---------------|------------|--------|
| LoftQ | LlaMA-3-8B | $5 \times 10^{-5}$ | 8 | 3 |
| SpaLLM | LlaMA-3-8B | $1 \times 10^{-5}$ | 8 | 3 |

## B.3  FINE-TUNING ON LLaMA-7B

We fine-tuned LlaMA-7B using SpaLLM on the entire Alpaca dataset, applying the standard prompt as described in the official Alpaca dataset repository (Taori et al., 2023). We trained 10 epochs on the dataset with a batch size of 16 and a learning rate of $3 \times 10^{-5}$. We selected the learning rates from ($3 \times 10^{-5}$, $1 \times 10^{-5}$, $3 \times 10^{-6}$, $1 \times 10^{-6}$).

Table 9: An example of LLM-as-a-judge comparison on model generation using LoftQ and SpaLLM on LLaMA-3-8B. In this example, SpaLLM is given to the GPT-4 judge as "Model 1," while LoftQ is "Model 2." The judge conducts two separate evaluations: one with Model 1's output appears before Model 2's, and another where Model 2's output presented first. In the first evaluation, the judge concludes that Assistant A (Model 1) performs better. In the second evaluation, following the change in output order, the judge again determines that Assistant B (which, due to the swap in order, corresponds to Model 1) performs better.

| Input: | Below is an instruction that describes a task. Write a response that appropriately completes the request.\n\n### Instruction:\nExplain what foreign direct investment (FDI) is.\n\n### Response:\n |
|---|---|
| Model | Output |
| SpaLLM (model 1) | "Foreign direct investment (FDI) refers to the investment made by a company from one country into a business or assets in another country. It involves the direct purchase of ownership or control over the assets or operations of a company in a foreign country. FDI can take many forms, such as the acquisition of existing businesses, the establishment of new facilities or operations, or the purchase of assets such as real estate or equipment. FDI can be beneficial for both the investing company and the host country, as it can lead to increased economic activity, job creation, and technological transfer. However, it can also pose challenges for host countries, such as the loss of control over their industries or the impact on their labor markets. As such, governments often regulate FDI to ensure that it is carried out in a way that benefits the host country and its citizens." |
| LoftQ (model 2) | "Foreign direct investment (FDI) is when a company or individual from one country invests money in a business in another country. It is a type of investment that allows the investor to directly own and control a business in a foreign country." |
| GPT-4 Judgment | |
| Judgment 1 | "Assistant A provides a more comprehensive and detailed explanation of foreign direct investment (FDI). It covers various aspects such as the forms FDI can take, the benefits and challenges associated with it, and the role of government regulation. This depth of information makes the response more helpful and relevant to someone seeking to understand FDI.\n\nAssistant B's response, while accurate, is very brief and lacks the depth and detail found in Assistant A's response. It provides a basic definition but does not elaborate on the different forms of FDI, its potential benefits and challenges, or the regulatory aspects.\n\nTherefore, Assistant A's response is more thorough and informative.\n\n[[A]]" |
| Judgment 2 | "Assistant A provides a concise and accurate definition of foreign direct investment (FDI), explaining that it involves a company or individual from one country investing in a business in another country, with direct ownership and control.\n\ Assistant B offers a more detailed explanation, covering various forms of FDI, potential benefits such as economic activity and job creation, and challenges like loss of control over industries. Additionally, Assistant B mentions the regulatory aspect of FDI by governments.\n\nWhile both responses are accurate, Assistant B's response is more comprehensive, providing a deeper understanding of FDI, its forms, benefits, challenges, and regulatory considerations.\n\ Therefore, the better response is:\n\n[[B]]" |

### B.4 FINE-TUNING ON LLAMA-3-70B

We used the entire Alpaca dataset as our training data. Due to the large size of the dataset and the base model, we trained one epoch on the dataset, and used the last checkpoint for evaluations. We tried learning rates from $(3 \times 10^{-5}, 1 \times 10^{-5}, 3 \times 10^{-6}, 1 \times 10^{-6})$, and used $1 \times 10^{-5}$ as our final learning rate.

