# OpenReview forum: "SpaLLM: Unified Compressive Adaptation of Large Language Models with Sketching"
_ICLR.cc/2025/Conference — Submitted to ICLR 2025_

### Official Review · Reviewer_Bvwr · 2024-10-30

**Soundness:** 3
**Presentation:** 3
**Contribution:** 3
**Rating:** 5
**Confidence:** 4

**Summary:**

The paper introduced a novel compressive adaptation approach (SpaLLM) for LLMs that effectively addresses the limitations of existing methods QLoRA. Spallm maintains high performance while reducing computational and memory usage.

**Strengths:**

1. The paper proposed sketched parameters, an interesting idea for compressive adaptation.

2. The paper is well written. the proposed approach is simple and clear.

**Weaknesses:**

1. In section 2.3. how did you fix the mapping introduced by the sketching matrix?
2. in all the experimental settings, you choose the Lora rank equal to 64. I am not sure this comparison is fair. Could you show the results when Lora's rank equals 4 or 1?
3. Why did you choose 5 common sense QA datasets in Table 3 while 7 common sense qa datasets in Table 4? Could you make it consistent?
4. in Figure 3, the accuracy is still improving as the increase of training parameters. Could you show more training parameters? in other words, is it scaling to more training parameters?

5. There is no code for the paper.

I promise I would like to raise my score if these questions can be solved.

**Questions:**

Please refer to the weakness.

---

> ### Author Response · Authors · 2024-12-03
>
> We are thankful for the reviewer's close reading and helpful feedback. The following are our responses to the points raised.
>
> > **[W1] Fixing the Mapping Introduced by the Sketching Matrix**
>
> We address the mapping introduced by the sketching matrix using an iterative rounding process. Specifically, we iteratively round each parameter in $\theta$ to the nearest centroid in $w$, while updating the not-yet-rounded parameters in $\theta$ to compensate for errors introduced by rounding. This approach follows the update rule described in Frantar et al. and is detailed on line 190 in the paper.
>
> > **[W2] Choice of LoRA Rank (Rank = 64)**
>
> The choice of a fixed LoRA rank of 64 is based on the LoftQ paper, which conducted extensive hyperparameter searches and determined that a rank of 64 achieves the best performance for LoRA, QLoRA, and LoftQ. To ensure consistency and fairness, we aligned our comparisons with their findings.
>
> > **[W3] Consistency in Commonsense QA Dataset Selection**
>
> We thank the reviewer for bringing this to our attention. To address this, we will conduct additional experiments to unify the dataset selection and include the updated results in the camera-ready version of the paper.
>
> > **[W4] Scaling to More Training Parameters**
>
> We have demonstrated the accuracy improvement as the number of training parameters scales up in Figure 3 of the paper. We will add further experiments with a larger range of training parameters in the camera-ready version.
>
> > **[W5] Code Availability**
>
> We will release the code upon acceptance of the paper.

---

### Official Review · Reviewer_XrAo · 2024-11-01

**Soundness:** 2
**Presentation:** 2
**Contribution:** 2
**Rating:** 3
**Confidence:** 4

**Summary:**

The paper presents a unified approach to compressive adaptation of LLMs, namely SpaLLM. The method allows adaptation without the strict low-rank matrix assumption. The authors claim that the method is more efficient and have better performance than methods like QLoRA. A row-wise parameter sketching method is introduced to skech the parameters for fine-tuning. It supports unified compressive adaptation with a single-tower architecture, task-specific compression and inherently supoorts the dynamic management of multiple adapters. The method is compared with several popular compressive LoRA methods and demonstrate comparable efficiency and accuracy.

**Strengths:**

1. The method demonstrates precision improvements on sevaral LLM finetuning benchmarks, which is a hot topic in the literature.
2. The approach simplifies LLM's compressive adaptation workflow.

**Weaknesses:**

1. The technical contribution seems limited and is not well elaborated. I expect some formal mathematical specification of the sketching method.The authors try to elaborate a lot on the use cases of the mehtod, however, the detailed description on how to implement those ideas are missing;
2. The evaluation in figure 3 shows that accuracy on llama2 7B have better accuracy when GPR=8, however, in other evaluations, GPR seems to be randomly setted to 1，4 ，8.  It makes the readers confused on the selection of GPR even with an ablation study. Moreover, the efficiency is only compared when GPR=1 in Table 5, which may lead to poteintial decay on the model performance.
3.On the settings of experiments, it seems that the authors carefully selects some good results to present without introducing the intuition on the selection of certain parameters.

**Questions:**

1. The authors listed some compressive adapation methods like DoRA, OFT and BOFT. Did the author try to make exprimental comparison with these methods?

---

> ### Author Response · Authors · 2024-12-03
>
> We greatly appreciate the reviewer’s thorough evaluation and constructive feedback. Please find our responses to your questions below.
>
> > **[Q1]: Comparison with DoRA and Other Methods**
>
> We compare our proposed method, SpaLLM, with QDoRA (DoRA used as adapters for QLoRA fine-tuning). The results for QDoRA, QLoRA, full fine-tuning, five-shot, and zero-shot settings are obtained from the DoRA paper [1]. For our experiments, we fine-tune SpaLLM LLaMA-2-7B on the Orca-Math dataset, and the results are presented in the table below.
>
> | Method         | #Bits | Accuracy (%) |
> |----------------|-------|--------------|
> | SpaLLM (GPR=4) | 4     | **0.34**     |
> | QDoRA          | 4     | 0.31         |
> | QLoRA          | 4     | 0.12         |
> | Full Fine-tune | 16    | 0.26         |
> | Five-shot      | 16    | 0.08         |
> | Zero-shot      | 16    | 0.07         |
>
> The Orca-Math dataset contains 200k training samples. Following the experimental setup outlined in QDoRA’s evaluation, we use 100k samples for training and 500 samples for evaluation. Accuracy is measured using the exact match score.
>
> As shown in the table, SpaLLM significantly outperforms QLoRA. Furthermore, SpaLLM achieves higher accuracy compared to both QDoRA and full fine-tuning, highlighting its ability to generalize effectively to diverse tasks while consistently surpassing existing compressive adaptation methods.
>
> References
>
> [1] Liu, Shih-yang, et al. "DoRA: Weight-Decomposed Low-Rank Adaptation." Forty-first International Conference on Machine Learning.

---

### Official Review · Reviewer_4wSV · 2024-11-03

**Soundness:** 2
**Presentation:** 1
**Contribution:** 2
**Rating:** 1
**Confidence:** 4

**Summary:**

The paper introduces a parameter-efficient finetuning (PEFT) method called SpaLLM for the compressive adaption of LLMs. It investigates the assumption and overhead issues of “two-tower” QLoRA-like PEFT methods. SpaLLM sketches the LLM weights into lookup tables and finetunes the table values without the low-rank assumption. The paper claims SpaLLM provides better accuracy and efficiency tradeoffs than previous methods.

**Strengths:**

- The paper presents an interesting PEFT method that adapts LLM model weights by sketching without relying on LoRA style low-rank adaption techniques.

- The experiments show competitive results than previous PEFT methods

**Weaknesses:**

- The paper is potentially flawed without further clarification on the sketching technique. The introduction and method sections are vague, unclear, and incoherent.

Examples include:
1. The two-tower architecture is defined without any context, it normally means two branches of some encoder-like architecture, like in the Siamese network. From figure 2, the paper seems to indicate the weights getting finetuned in separate two steps and finetuning works like a two-twoer style.
2. In the introduction, it is unclear how QLoRA causes a performance gap and how LoftQ reduces such a gap. And what are those performance gaps, task accuracy loss, or efficiency gaps? The introduction says the compression process is lossy (line 63), but why is it lossy? And why is the sketching via parameter sharing not lossy? Regarding difficulty in implementation, isn’t there a QLoRA library that makes it easy for people to use, the paper also claims QLoRA has additional overhead and requires significant system-level optimizations (line 73), but it does not get quantified.
3. The presentation of SpaLLM and sketching ideas in the introduction are vague and redundant and do not contain much useful information. Line 78 says finetuning directly on compressed model parameters, how are the parameters compressed? Seems the parameters are quantized to low-bit precision and then applied to sketching. Sketching is parameter sharing and not contradictory compression. Line 85-94 then discuss the look-up tables (LUTs), how is sketching connected to these LUTs is unknown. How the LUTs are used also remains unclear. The introduction has no results or any comparison with previous methods.

4. Section 2.1 seems like a related work section but turns out not since it is mixed with existing work and the SpaLLM sketching-based parameter sharing. Line 113 - 122 says the two virtual and trainable parameter categorizations, which seems unnecessary because they are not referred to anywhere later in the paper.

5. The Sketching idea is in fact parameter sharing from section 2.2, which is no new. The paper repurposes the parameter sharing for PEFT. However, it is unclear how the sketching works,  from lines 148 - 151, it consists of row-wise parameter sketching, weighted Lloyd optimization, and LUTs. It does not explain how these three components work together and why they are needed. The weighted Lloyd algorithm seems to avoid collision during hashing, so it is confusing to rebrand the hashing as sketching.

- While the results are strong and much better than the compared baselines, without knowing the details of the SpaLLM method makes it much less convincing and difficult to evaluate the true improvements.

**Questions:**

- Is the centroid learning process clustering?  are all of this Lloyd algorithm computation done during training? the overhead seems huge
- The one-hot encoding does not save memory, any compression of one-hot encoding?

- Line 196, what is the role of sketched parameters? is it similar to LoRA ranks? or just tuning more parameters? What are groups per row?

- Line 206, why do the sketch parameters become floating-point? which set of parameters are fixed-point?

- Line 215, no explanation of LUTs, how do you build it, how to maintain it in GPU memory? extra overhead?

- Line 241, why only the sketched parameters are stored and updated for each task, not the LUTs? Do you still need LUTs for inference?

- Line 255, how would compression not lose information? the identify matrix example is not convincing because real model weights are not identity.

---

> ### Author Response · Authors · 2024-12-03
> **Response 1 of 2**
>
> We appreciate the reviewer's thorough review and valuable feedback. Below, we address your questions in detail.
>
> > **[Q1]: Centroid Learning Process & Overhead**
>
> We use Lloyd’s algorithm for learning the LUTs, which is a form of clustering. The overhead is lightweight, as building the LUT only requires 1 hour for an 8B model and 6 hours for a 70B model. We will add more details for centroid learning overhead in the final paper.
>
> > **[Q2]: One-hot Encoding**
>
> We do not use one-hot encoding in our implementation; it is only presented in the paper to illustrate the concept. In practice, the one-hot representation is stored as indices, which are encoded as low-bit integers on GPUs, ensuring efficient memory usage.
>
> > **[Q3]: Line 196 - Role of Sketched Parameters**
>
> The sketched parameters serve to capture the knowledge contained in both the original LLM and the fine-tuned LLM. While they are conceptually similar to the parameters of low-rank adapters, a key difference lies in initialization: low-rank adapters are typically zero-initialized, whereas the sketched parameters already encode the knowledge from the original LLM. This dual role allows the sketched parameters to function both as the model parameters and as the adapter for fine-tuning.
>
> Regarding "groups per row," this refers to the number of Look-Up Tables (LUTs) assigned to each row in the original weight matrix. Increasing the number of groups per row provides more trainable parameters, enhancing the model's capacity for fine-tuning.
>
> > **[Q4]: Line 206 - Number Format of Sketched Parameters**
>
> We would like to direct the reviewer’s attention to Figure 1, where we illustrate how sketch parameters are stored in our model. The sketched parameters are stored as floating-point numbers to enable differentiable training and to maintain the precision of the sketched model. These floating-point values are stored in the LUTs. In contrast, the keys used to index into the LUTs are represented as low-bit integer codes, which are frozen and do not take gradients. This design ensures efficient memory usage while preserving the model's trainability and accuracy.
>
> > **[Q5]: Line 215 - Process of Building LUTs and Overheads**
>
> We appreciate the reviewer’s observation and provide the following details regarding Look-Up Tables (LUTs):
>
> **Building the LUTs:** To construct the LUTs, we first perform Lloyd’s algorithm (Lloyd, 1982) to learn a set of $k$ centroids for the rows of parameters, with the parameters inversely weighted by the Hessian diagonals. This ensures the centroids are representative of the underlying parameter distributions while prioritizing more impactful parameters.
> Once the centroids $w$ are established, we learn the sketching matrix using the iterative loss-error-based quantization framework (Zhang & Shrivastava, 2024). Specifically, each parameter in $\theta$ is iteratively rounded to the nearest centroid in $w$, and the not-yet-rounded parameters in $\theta$ are updated to compensate for rounding errors. This process follows the update rule described in Frantar et al. (2022).
>
> **Maintaining LUTs in GPU Memory:** The LUTs are held in shared memory on the GPU to enable fast access and lookups. This approach ensures low latency overhead during training and inference while leveraging the high-speed memory architecture of GPUs for efficient computations.
>
> **Extra Overhead:** The overhead introduced by the LUTs is minimal because the number of centroids is much smaller than the number of original parameters. Additionally, the iterative quantization framework ensures that the LUTs effectively approximate the original parameters without a significant loss in accuracy.
>
> We will include a detailed explanation of LUT construction, maintenance, and overhead in the final version of the paper.
>
> > **[Q6]: Line 241 - Sketched Parameters vs. LUTs in Task Updates and Inference**
>
> The sketched parameters are stored as values within the LUTs. When a compressed model is adapted to a downstream task using SpaLLM, the trained LUTs containing these sketched parameters are saved. During inference, these LUTs are loaded into memory, enabling the compressed model to use low-bit integer codes as keys to retrieve the adapted sketched parameters.
> This approach allows SpaLLM to scale efficiently for multi-task training and inference. The compressed model remains consistent in memory, while separate LUTs for each task are loaded and updated independently. This design supports parallel task adaptation and efficient deployment across diverse tasks.

---

> > ### Author Response · Authors · 2024-12-03
> > **Response 2 of 2**
> >
> > > **[Q7]: Line 255 - Maintaining Information Integrity in Compression**
> >
> > We acknowledge the limitations of the identity matrix example as a theoretical abstraction. Since the identity matrix has only two distinct values, it can be losslessly compressed using a simple LUT of two values (0 and 1). In contrast, real-world weight distributions are more complex. Our method addresses this complexity, as evidenced by the experimental results, which demonstrate its ability to preserve accuracy across diverse models and tasks, even with non-identity weight structures.

---

### Official Review · Reviewer_c5gG · 2024-11-04

**Soundness:** 3
**Presentation:** 3
**Contribution:** 3
**Rating:** 6
**Confidence:** 4

**Summary:**

SpaLLM (Sketched Parameter Adaptation of LLMs), a novel approach for compressive adaptation of large language models (LLMs) that simplifies the fine-tuning process while enhancing efficiency and accuracy. Unlike traditional methods such as QLoRA, which use a "two-tower" architecture with low-rank assumptions, SpaLLM employs a sketching-based parameter-sharing technique that avoids these constraints. This unified approach combines model compression and adaptation into a single, streamlined process.

Key contributions of the paper include:

* Single-Tower Architecture: SpaLLM eliminates the need for the two-tower structure, reducing computational overhead by requiring only one compressed matrix multiplication per layer during inference.
* Sketching-Based Compression: The method applies row-wise parameter sketching with optimized mappings, allowing direct adaptation of compressed parameters without relying on low-rank algebraic assumptions.
* Improved Efficiency and Accuracy: Experimental results show that SpaLLM outperforms state-of-the-art compressive adaptation methods in both natural language understanding and generation tasks, delivering significant efficiency gains in memory and inference speed.
* Task Flexibility: The approach supports dynamic maintenance of multiple adapters, allowing seamless task-switching and adaptability for multi-user serving environments.

SpaLLM offers a resource-efficient solution for fine-tuning large models, making it particularly suited for deployment on hardware with limited resources.

**Strengths:**

The paper presents an innovative solution to compressive adaptation by introducing SpaLLM, which avoids the traditional limitations of low-rank, two-tower architectures seen in methods like QLoRA. By employing a sketching-based parameter-sharing approach, it redefines how parameter compression and adaptation can be unified. This approach not only removes algebraic constraints but also opens new avenues for efficient fine-tuning, making SpaLLM a notable contribution that combines original ideas with practical advancements in LLM adaptation.

## Quality
The quality of the work is high, demonstrated through meticulous experimentation and comparison against established state-of-the-art methods. The experiments cover various LLM architectures and datasets, including LLaMA and CommonsenseQA benchmarks, highlighting the method’s accuracy and efficiency. The authors rigorously test SpaLLM’s performance under different bit compression rates, group sizes, and model configurations, showcasing the robustness of the approach and supporting the claims with thorough empirical evidence.

## Clarity
The paper is generally well-organized, with each section logically leading to the next. The authors provide clear definitions and motivations for concepts such as "two-tower architectures" and "parameter-sharing compression." Diagrams and tables, such as the illustration of row-wise parameter sketching and comparisons with baselines like LoftQ, further enhance clarity by visually presenting complex ideas. Although highly technical in parts, the paper effectively communicates its contributions and methodology.

## Significance
SpaLLM has high practical and theoretical significance, especially given the growing need for resource-efficient LLM adaptation techniques. By enabling adaptation directly on compressed parameters, SpaLLM reduces computational and memory overhead, which has profound implications for deploying LLMs on hardware with limited resources. Its flexibility in managing multiple adapters and dynamic task-switching makes it particularly valuable for multi-task learning scenarios, enhancing its potential for broader, real-world applications in natural language understanding and generation.

Overall, the paper’s strengths lie in its innovative approach to compressive adaptation, robust methodology, clear presentation, and significant contributions to scalable and efficient LLM adaptation.

**Weaknesses:**

In general , I think this is a great work.

Some concerns are:

* the success of SpaLLM relies heavily on the effectiveness of its sketching algorithm. If the sketching fails to capture essential details of the parameter distributions, it can lead to a loss in model accuracy or even failure to converge, particularly under aggressive compression settings.

* SpaLLM’s parameter-sharing approach could introduce challenges in tasks that require highly specialized or fine-grained learning (e.g., nuanced natural language understanding or highly specific domain tasks), as parameter-sharing might restrict the model’s flexibility.

**Questions:**

How to demonstrate the effectiveness of the sketching algorithm.

---

> ### Author Response · Authors · 2024-12-03
>
> We sincerely thank the reviewer for recognizing the significance of our work and for providing constructive feedback. Below, we address the concerns raised:
>
> > **[W1] Effectiveness of the Sketching Algorithm**
>
> We acknowledge the reviewer's concern regarding the reliance of SpaLLM on the sketching algorithm. We highlight that, in our experiments, we rigorously evaluated SpaLLM's sketching capability across multiple state-of-the-art LLMs, including LLaMA-7B, LLaMA-2-7B, LLaMA-2-13B, LLaMA-3-8B, and LLaMA-3-70B. The results consistently demonstrate that our sketching algorithm effectively captures the essential parameter distributions of these models, even under aggressive compression settings.
>
> > **[W2] Generalization Across Diverse Tasks**
>
> While parameter-sharing might raise concerns for tasks requiring specialized learning, our empirical results substantiate SpaLLM's flexibility and effectiveness across a diverse set of benchmarks, including language modeling (WikiText-2), math (GSM8K), instruction tuning (Alpaca), and commonsense reasoning (CommonsenseQA).

---

### Meta-Review · Area_Chair_tq2v · 2024-12-21

**Metareview:**

The submitted paper presents a novel approach for enhancing model efficiency through the use of Look-Up Tables and sketched parameters, aiming to optimize the training process for large language models. The authors argue that this idea is significant as it addresses current challenges in memory usage and computation efficiency in AI models, potentially impacting various applications in machine learning.

The reviewers shared mixed opinions about the paper. While one reviewer strongly rejected the submission, citing issues with the methodology and insufficient justification of the presented models, others acknowledged its potential and provided constructive feedback. This discrepancy in perspectives highlights a lack of consensus among the reviewers regarding the paper's contributions and overall impact.

Despite the authors' efforts to address the reviewers' concerns in their rebuttal, several critical issues remain unresolved. The reviewers still expressed doubts about the effectiveness of the identity matrix example and the overhead of the lookup tables. Additionally, the reviewers were concerned that the clarity on the mathematical underpinnings and the integration of sketched parameters in practical scenarios are inadequate.

Given the unresolved concerns post-rebuttal, it is recommended that this paper be rejected. The authors should consider addressing the highlighted issues comprehensively in future submissions.

**Additional Comments On Reviewer Discussion:**

The reviewers shared mixed opinions about the paper. While one reviewer strongly rejected the submission, citing issues with the methodology and insufficient justification of the presented models, others acknowledged its potential and provided constructive feedback. This discrepancy in perspectives highlights a lack of consensus among the reviewers regarding the paper's contributions and overall impact.

---

### Decision · Program_Chairs · 2025-01-22

Reject